# Predicting the Pathway Involvement of All Pathway and Associated Compound Entries Defined in the Kyoto Encyclopedia of Genes and Genomes

**DOI:** 10.3390/metabo14110582

**Published:** 2024-10-27

**Authors:** Erik D. Huckvale, Hunter N. B. Moseley

**Affiliations:** 1Markey Cancer Center, University of Kentucky, Lexington, KY 40536, USA; edhu227@uky.edu; 2Superfund Research Center, University of Kentucky, Lexington, KY 40536, USA; 3Department of Toxicology and Cancer Biology, University of Kentucky, Lexington, KY 40536, USA; 4Department of Molecular and Cellular Biochemistry, University of Kentucky, Lexington, KY 40536, USA; 5Institute for Biomedical Informatics, University of Kentucky, Lexington, KY 40536, USA

**Keywords:** pathway prediction, Matthews correlation coefficient, machine learning, multi-layer perceptron, transfer learning, KEGG

## Abstract

**Background/Objectives**: Predicting the biochemical pathway involvement of a compound could facilitate the interpretation of biological and biomedical research. Prior prediction approaches have largely focused on metabolism, training machine learning models to solely predict based on metabolic pathways. However, there are many other types of pathways in cells and organisms that are of interest to biologists. **Methods**: While several publications have made use of the metabolites and metabolic pathways available in the Kyoto Encyclopedia of Genes and Genomes (KEGG), we downloaded all the compound entries with pathway annotations available in the KEGG. From these data, we constructed a dataset where each entry contained features representing compounds combined with features representing pathways, followed by a binary label indicating whether the given compound is associated with the given pathway. We trained multi-layer perceptron binary classifiers on variations of this dataset. **Results**: The models trained on 6485 KEGG compounds and 502 pathways scored an overall mean Matthews correlation coefficient (MCC) performance of 0.847, a median MCC of 0.848, and a standard deviation of 0.0098. **Conclusions**: This performance on all 502 KEGG pathways represents a roughly 6% improvement over the performance of models trained on only the 184 KEGG metabolic pathways, which had a mean MCC of 0.800 and a standard deviation of 0.021. These results demonstrate the capability to effectively predict biochemical pathways in general, in addition to those specifically related to metabolism. Moreover, the improvement in the performance demonstrates additional transfer learning with the inclusion of non-metabolic pathways.

## 1. Introduction

A wide variety of small biomolecules are found in living systems and are involved across all biological processes. Most of these biomolecules are involved in biochemical reactions that comprise cellular metabolism, which is typically organized into metabolic “pathways”, a term that may refer to classical pathways, for example glycolysis, or larger pathway categories (sometimes called modules), for example carbohydrate metabolism. These pathways form a network of metabolites interconnected by biochemical reactions that can be represented as graphs (technically requiring hypergraphs), where the (hyper) edges are reactions and the nodes are metabolites involved in a given reaction [1,2,3]. Several knowledge bases, such as the Kyoto Encyclopedia of Genes and Genomes (KEGG) [4,5,6], MetaCyc [7], and Reactome [8], contain pathway annotations for many biomolecules. This manually defined, pathway-level organization of biomolecules is highly useful for interpreting molecular experimental data derived from a biological system. However, knowledge of the pathway involvement of biomolecules is incomplete, and these knowledge bases are missing many biomolecules and associated pathway annotations, many likely due to unknown enzyme promiscuity. Moreover, it is costly, time consuming, and tedious to generate and interpret experimental data to derive new pathway annotations, requiring specialized analytical and biochemical expertise.

In response to these limitations, there is strong interest in developing alternative methods that can provide pathway annotations for detected biomolecules. In particular, machine learning models can be trained to predict the pathway involvement of compounds lacking pathway annotations using compounds with known annotation. Toward this end, Huckvale et al. created a KEGG-based benchmark dataset for metabolism [9], generating features representing compounds using an atom coloring technique developed by Jin and Moseley [10]. KEGG organizes its pathways in a hierarchy where higher levels in the hierarchy are categories of pathways and the lowest level of the hierarchy comprises individual pathways: https://www.genome.jp/kegg-bin/show_brite?br08901.keg (accessed on 1 June 2024). Included in Level 1 (L1) of the hierarchy is the “Metabolism” top category, with 12 Level 2 (L2) subcategories related to metabolism. The models trained on the benchmark dataset, as well as the models from past publications, were designed to predict the associations with only these L2 metabolic pathways, with many of the models limited to predictions for only 11 of the 12 L2 “Metabolism” pathways [11,12,13,14,15]. These models were severely limited by pathway granularity due to the size limitations of the training dataset. Using a modification of this benchmark dataset where both compound and pathway features were generated, Huckvale and Moseley demonstrated the capability of training a single binary classifier that accepts a compound representation and a pathway representation and outputs whether the given compound is associated with the given pathway [16]. The cross-join of metabolite and pathway entries increased the size of the training set from ~5600 to nearly 70,000 entries. Since this model was capable of predicting an arbitrary number of pathways, Huckvale and Moseley expanded it to predict not only the L2 metabolic pathways but also the Level 3 (L3) individual pathways, improving the performance when the dataset increased to over 1,000,000 entries [17]. However, it was still restricted to pathways below the L1 “Metabolism” pathway category, while there are several other types of pathways of interest to biologists, including “Human Diseases” and “Genetic Information Processing”. In this work, we expand on our prior work to include all the pathways in the KEGG hierarchy (L1, L2, and L3) and all the KEGG compounds with any pathway annotation, creating a dataset with over 3,200,000 entries. The resulting binary classifier can predict the association of compounds to any KEGG pathway, not just metabolic pathways, with improved prediction performance.

## 2. Materials and Methods

The KEGG organizes its pathways in a hierarchy with three levels, which is found here: https://www.genome.jp/kegg-bin/show_brite?br08901.keg (accessed on 1 June 2024). The top level of the hierarchy contains the top pathway categories, which we will call L1 pathways. The next level contains the pathway categories or modules, which we will call L2 pathways. The lowest level contains the individual human-defined pathways, which we will call L3 pathways. The L1 pathways include “Metabolism”, “Genetic Information Processing”, “Environmental Information Processing”, “Cellular Processes”, “Organismal Systems”, “Human Diseases”, and “Drug Development”. While past publications have only considered the L2 under “Metabolism” and recently the L3 metabolic pathways, we used the kegg_pull [18] Python package to download all the pathways in the hierarchy, including the L1 pathways and all the L2 and L3 pathways underneith them. We additionally used kegg_pull to download the KEGG compounds as molfiles and determined the mapping from the compound to the associated pathways using the pathway annotations provided by the KEGG.

With the molfiles and pathway mappings, we used the dataset construction method described by Huckvale and Moseley [16]. This includes converting the molfiles to compound features using the atom coloring technique introduced by Jin and Moseley [10], constructing the pathway features based on the features of the compounds associated with them, and performing a cross-join of the compound features and the pathway features where each entry in the resulting dataset is a pair of pathway features and compound features concatenated together and the binary label indicates whether the corresponding compound is associated with the corresponding pathway. In order to maximize the validity of the evaluation of the test sets in the cross-validation (CV) analysis (i.e., prevent data leakage from the training sets into the test sets), any duplicate compound or pathway feature vectors were removed prior to the cross-join, resulting in removing 20 duplicate pathway entries and 97 compound entries. Considering the significantly larger amount of data, we were motivated to make the data loading technique used in model training more efficient. While both this work and the work of Huckvale and Moseley [17] uses the PyTorch Python package [19] for implementing and training the multi-layer perceptron (MLP) binary classifier, we did not use PyTorch’s built-in data loader in this work. Previously, the built-in data loader was used, retrieving each entry one at a time by selecting the current compound feature vector and pathway feature vector and concatenating them together, followed by inserting the resulting vector into the next batch and loading the batch onto the graphics processing unit (GPU). This batching technique was used because the entire dataset could not fit into the available GPU random access memory (RAM) all at once. While the cross-joined dataset was not able to fit in the GPU, the separated compound features and pathway features could. This enabled us to create our own data loader that samples the compound features and pathway features in batches rather than one at a time and concatenates them together on the GPU. With all the data having been loaded onto the GPU ahead of time and all the batching being performed by efficient tensor function calls on the GPU rather than numpy function calls on the central processing unit (CPU), we were able to reduce the training time of the model by more than 20-fold. Appendix A shows the difference in computational resource usage between training the final model on the full KEGG dataset using the previous data loading method and that using our novel data loading method. We see a stark increase in GPU utilization usage from 8.8% to 93.7%, followed by a stark decrease in the real computation time from 978.2 min to 47.1 min. The speed improvement is over 20-fold, even though the increase in GPU utilization is only 10.6-fold. We suspect that the custom data loader avoids costly GPU wait states with the transfer of data from the CPU RAM to the GPU RAM, providing better efficiency than one would first expect. These results demonstrate that making better use of the GPU in the data loading greatly decreases the model training time.

After constructing the dataset and implementing the novel data loading technique, we tuned the model hyperparameters using the Optuna Python package version 3.3.0 [20]. With the best hyperparameters selected, we performed an analysis of 200 CV iterations on the entire dataset using stratified train/test splits [21]. In these CV analyses, we tracked not only the total number of true positives, true negatives, false positives, and false negatives of all the dataset entries but those of each individual compound and pathway as well. This enabled us to not only construct a confusion matrix and calculate overall metric, but also to calculate the metrics per compound and per pathway as well. The metrics calculated included the Matthew’s correlation coefficient (MCC), accuracy, precision, recall, F1 score, and specificity. In order to ensure valid (i.e., not undefined due to division by 0) metrics for each compound and pathway, we constructed the confusion matrix of each by summing the true positives, true negatives, false positives, and false negatives across all CV iterations. This manner of calculation prevents obtaining a standard deviation. So, for the full dataset, we calculated the metrics per CV iteration and calculated the mean, median, and standard deviation across the CV iterations.

To determine the impact of the chemical information content of the compounds and pathways on the model performance, we additionally created filtered datasets (from a preliminary dataset constructed prior to de-duplicating the pathway and compound entries). First, we created 15 datasets, each with an increasingly higher compound filter threshold. The filtering was based on the number of non-hydrogen atoms in a compound and compounds were removed from the training set if the number of non-hydrogen atoms did not meet each filter threshold. Appendix A shows how the number of compounds and entries in the dataset decreases as the compound filter cutoff increases.

We performed a CV analysis of 50 CV iteration on each filtered dataset. We then filtered by pathway size, defining the pathway size as the sum of the number of non-hydrogen atoms across all the compounds associated with the pathway. The pathway filters first ranged from 5 to 50 by multiples of 5, 50 to 100 by multiples of 10, and then 100 to 200 by multiples of 20, a total of 20 pathway filters. We then performed 50 CV iterations on the training set of each pathway filter. The motivation was to determine the ideal compound size and pathway size for the full KEGG dataset. Appendix A shows how the number of pathways and entries in the dataset changes as the pathway filter increases.

Appendix A shows scatter plots of the thresholds used to filter each training set (see Appendix A) and compares the thresholds to the MCCs of the top compounds in Appendix A and the top pathways in Appendix A. For consistent comparison, the top compounds are the compounds remaining in the dataset of the highest compound size filter threshold (i.e., 15) and the top pathways are the pathways remaining after the highest pathway filter threshold (i.e., 200). This is because we cannot justify removing data from the dataset merely because the overall score is higher, since this can only occur because the smaller compounds and pathways are removed and they may be more difficult to predict. But if the smaller compounds and pathways negatively impact the larger compounds and pathways, then it is best to remove them. However, Table 1 shows that these Pearson correlation coefficients are very close to zero, even though their *p*-value are statistically significant. Thus, these relationships are real, but they are very weak. Due to these negligible correlations, we decided to retain all the compounds and pathways for the final model training and evaluation. The data and results of this preliminary analysis can be found in the following Figshare: https://figshare.com/articles/journal_contribution/FullKEGG_Preliminary_DO_NOT_USE/27173037 (accessed on 4 October 2024).

In addition to testing the impact of filtering entries by compound and pathway size, we also tested filtering by hierarchy level. Table 2 shows how the number of entries and number of pathways differ between the full dataset containing the L1, L2, and L3 pathways and two other datasets i.e., that containing only L2 and L3 pathways, and lastly, that containing L3 pathways only. The number of compounds remains the same regardless of which pathway hierarchy levels are included. We trained on each of these three datasets to test how the inclusion of one hierarchy level impacts the performance of the others. The L2 and L3, as well as the L3 only, training sets were evaluated over 50 CV iterations, with the metrics calculated by constructing a confusion matrix by summing the true positives, true negatives, false positives, and false negatives across all the included pathways across all CV iterations.

The hardware used for this work included machines with up to 2 terabytes (TB) of random-access memory (RAM) and central processing units (CPUs) of 3.8 gigahertz (GHz) of processing speed. The name of the CPU chip was “Intel(R) Xeon(R) Platinum 8480CL”. The CPUs were sourced from the Intel corporation in Santa Clara, California, USA. The graphic processing units (GPUs) used had 81.56 gigabytes (GB) of GPU RAM, with the name of the GPU card being “NVIDIA H100 80GB HBM3”. The GPUs were sourced from the Nvidia corporation in Santa Clara, California, USA.

All code for this work was written in the major version 3 of the Python programming language [22]. Data processing and storage were performed using the Pandas version 1.0.3 [23], NumPy version 1.26.4 [24], and H5Py version 3.9.0 [25] packages. Models were constructed and trained using the PyTorch Lightning package version 2.2.1 [26] built on the PyTorch package version 2.0.1 [19]. The metrics and the stratified train–test splits were computed using the Sci-Kit Learn package version 1.3.0 [27]. The results were stored in an SQL [28] database using the DuckDB package version 1.0.0 [29]. Data visualizations were produced using the Tableau business intelligence software version 2024.2.2 [30] as well as the seaborn package version 0.12.2 [31] built on the MatPlotLib package version 3.7.2 [32]. The results were finalized in a Jupyter notebook [33]. The computational resource usage when training the final model was collected using the gpu_tracker package version 3.0.0 [34]. All code and data for reproducing these analyses can be accessed via the following Figshare item: https://doi.org/10.6084/m9.figshare.27172941 (accessed on 21 October 2024).

## 3. Results

### 3.1. Main Results

The work of Huckvale and Moseley [17] produced the largest dataset for this task to date and it consisted of both the L2 and L3 pathways under the “Metabolism” L1 pathway with duplicate entries removed. However, our current work uses all the pathways in the KEGG hierarchy, including the L1 pathways not considered before (not just “Metabolism”) as well as all the L2 and L3 pathways underneath them. Table 3 shows the differences between the previous dataset (metabolic pathways) and that of this work (full KEGG).

Table 4 shows the mean, median, and standard deviation of the MCC calculated from all the predictions in each test set (all compound-pathway paired entries) across the 200 CV iterations. These aggregations of the 200 MCC scores are provided for each set of pathway hierarchy levels that were included in the dataset. This includes the L1, L2, and L3 pathways, which is the same dataset as the full KEGG in Table 1, followed by L2 and L3, and finally, L3 only. Appendix A contains these same scores for other metrics, including the accuracy, precision, recall, specificity, and F1 score.

Figure 1 shows the distribution of MCCs across CV iterations for each dataset. The L1, L2, and L3 dataset is the full dataset, containing all the pathways, and was run on 200 CV iterations. The L2 and L3 dataset excluded the L1 pathways and was run on 50 iterations. The L3 dataset contained only the L3 pathways and was also run on 50 CV iterations.

Table 5 shows the MCCs across the pathways of a certain hierarchy level for each set of pathway hierarchy levels included in the dataset. The MCCs were calculated by constructing a confusion matrix from the sum of the true positives, true negatives, false positives, and false negatives across all the pathways of the given hierarchy level across all CV iterations. For example, the L1 pathways in the full KEGG dataset scored an MCC of 0.950, while the L3 pathways, when trained on the dataset that only contains L3 pathways, scored an MCC of 0.726.

Table 6 shows the MCCs across the pathways under each L1 pathway when trained on the full KEGG dataset. The MCCs were calculated by summing the true positives (TPs), true negatives (TNs), false positives (FPs), and false negatives (FNs) of all the pathways under the L1 pathway, including the L1 pathway itself, and across all the CV iterations. We see that the pathways under “Genetic Information Processing” were the easiest to predict, while the pathways under “Cellular Processes” were the most difficult to predict.

Table 7 shows the MCC, F1 score, precision, recall, and specificity of the individual L1 pathways. We see that while “Genetic Information Processing” performed best when collectively predicting the pathways under it (Table 6), “Environmental Information Processing” performed best when predicting it by itself (Table 7). While “Metabolism” performed second best in Table 6, Table 7 shows that predicting whether a compound is a metabolite, i.e., associated with a metabolic pathway or not, is the most difficult. We see that for the other L1 pathways, the MCC is similar to the F1 score, which is typical. However, the MCC and F1 score are starkly different for “Metabolism”. We see that while precision and recall, which are based on positive predictions, are very high, the F1 score, which is based on precision and recall, is likewise high. Yet the specificity of “Metabolism”, which is based on negative predictions, is much lower, driving down the MCC. This demonstrates the superiority of the MCC as an overall performance metric, as it takes into account both positive and negative predictions; however, for certain applications, a more specific metric can be better.

Figure 2 provides an explanation for the discrepancy regarding “Metabolism” in Table 7. We see that “Metabolism” has a much larger size than the other L1 pathways (where the pathway size is defined by the total number of non-hydrogen atoms across all the compounds associated with the pathway), having more compounds associated with it and more positive entries in the dataset that correspond to the “Metabolism” pathway. The class imbalance problem [35] has made this machine learning task difficult due to the tendency of there being many more compounds that are not associated with a pathway while having relatively few that are associated with a pathway. However, the opposite but equally challenging problem of having too many positive entries exists for the “Metabolism” pathway, while the other pathways are challenged by having too many negatives. The true positives of “Metabolism” contribute to an improved F1 score, but the number of false negatives lowers the specificity. This also explains why the L1 pathways performed very well in Table 5, since that MCC benefitted from the high number of true positives in “Metabolism” combined with the plethora of true negatives in the remaining L1 pathways.

### 3.2. Comparing Pathway and Compound Size to MCC

Figure 3 shows the distribution of the size of the compounds (number of non-hydrogen atoms in the molecule) in the full KEGG dataset and that of the pathway size (total number of non-hydrogen atoms across the compounds associated with the pathway). To better see the peak of the pathway size distribution, Figure 3b shows only the pathway counts for those pathways with a size of 1000 or below, as compared to Figure 3a, which shows the distribution for all the pathways. The pathway that exceeds 160,000 in size is “Metabolism”, as shown in Figure 2.

Figure 4 shows the distribution of the MCC of individual compounds and pathways. We see that pathways center between 0.6 and 0.9 MCC, while others are closer to 0 and even slightly below 0 (i.e., slight inverse prediction). Even after 200 CV iterations, there were still four pathways without a valid MCC score, meaning we could not calculate the MCC without a division by zero, and therefore, they cannot be included in our results individually (their false negative and true negative counts do still contribute to the sums used to calculate the results in Table 5, Table 6 and Table 7). Appendix A shows these null pathways and their size. We see that the MCCs of the individual compounds are close to one and left skewed.

Figure 5 shows the relationship between the compound and pathway size and the respective MCC. When log scaling the *x*-axis, we see that there is not a strong linear correlation between the size and the MCC for either the pathways (Figure 5b) or the compounds (Figure 5d). However, we observe a funnel shape for the pathways such that there is less variance as the pathway size increases. And for the compounds, we see that the maximum compound size does not reach 1.0 until reaching a size of 5.

## 4. Discussion

Table 8 compares the results of training on all the KEGG pathways (Table 4) to those of training on only the metabolic pathways from the work of Huckvale and Moseley [17]. Specifically, Table 8 compares the mean and standard deviation of the MCCs of all the predictions in each test set across the CV iterations. The MCC of 0.847 for the overall performance of all the KEGG pathways compared to the MCC of 0.800 for that of the L2 and L3 metabolic pathways demonstrates a modest increase in performance when incorporating the L1 pathways and all the remaining L2 and L3 pathways under them.

Table 9 compares the results of training on all the KEGG pathways (Table 5) to the results of training on only the metabolic pathways from the work of Huckvale and Moseley [17]. Specifically, Table 9 compares the collective MCCs across the pathways of certain hierarchy levels separated by the hierarchy levels of the pathways included in the dataset used for training. We see that the performance of the L2 and L3 pathways remains comparable when adding both the L1 pathways and non-metabolic pathways to the dataset. The results of this work demonstrate the capability to not only effectively predict the metabolite association with metabolic pathways but also generic biomolecules with annotations to a broader set of biological and biomedical pathways.

From Table 6 and Table 7, we observe that certain L1 pathways are more difficult to predict than others, both when measuring the collective performance of the L1 pathway and all the L2 and L3 pathways under it as well as the individual performance of the L1 pathways alone. “Genetic Information Processing” performs best collectively, while “Environmental Information Processing” performs best individually. “Metabolism” performs more poorly individually, by a sizable margin. This is likely a result of having too many positive entries mapping to “Metabolism”, the opposite problem of having too few positives (and too many negatives), which has primarily been the challenge in predicting pathway involvement until now. This means that while we can effectively predict more specific metabolic pathways, effectively predicting whether or not a compound is a metabolite at all likely requires more compound entries associated with non-metabolic pathways. As demonstrated in Table 5, Table 8 and Table 9, these effects are partially ameliorated by transfer learning across the pathway hierarchy.

In a preliminary dataset in which we had not removed the duplicate pathway and compound feature vectors, CV analysis produced an average MCC of 0.822 (Appendix A). Upon removing the duplicate pathway and compound entries to maximize the validity of the CV analysis by preventing data leakage between the training and test sets, we observe an increase in the average MCC to 0.847 (Table 4). We believe that the inclusion of duplicate pathways and compounds added confusion to the training, which is suggested by the large drop in the standard deviation from 0.017 to 0.00098. While duplicate entries with conflicting ground truth (e.g., one feature vector corresponds to a positive label while a corresponding duplicate feature vector corresponds to a negative label) can lead to model confusion, we found that there were only 20 such entries in the dataset out of a dataset of over three million entries. So, a more plausible explanation for the added training confusion is that smaller compounds and pathways are more likely to have duplicate entries (duplicate atom color counts), while they are also more difficult to predict, so removing such entries can increase the overall MCC.

When including all the compounds and pathways available in KEGG in the training dataset, we do not observe a particularly strong linear correlation between the pathway and compound size and the MCCs of individual pathways and compounds. However, we still have evidence that both an increased compound size and an increased pathway size contribute to more reliable prediction. Specifically, there is less variance in performance in the case of pathways as the pathway size increases. And in the case of compounds, the maximum possible performance increases as the compound size increases.

## 5. Conclusions

While prior work on the machine learning task of predicting the pathway involvement of a compound has primarily dealt with metabolism, this work demonstrates that a model can be trained to effectively predict the pathway involvement of generic biomolecules with biological and biomedical pathways. Moreover, the prediction performance keeps improving as more compounds and pathways are included beyond merely metabolites and metabolic pathways. This marks a significant milestone in the field and we recommend that future work in this area should build on this standard. We believe that these models are demonstrating the level of performance needed for predicting pathway annotations for unannotated compound entries that are useful for certain application use-cases. One possible early use-case is providing hypotheses for unknown enzyme promiscuity.

## Figures and Tables

**Figure 1 metabolites-14-00582-f001:**
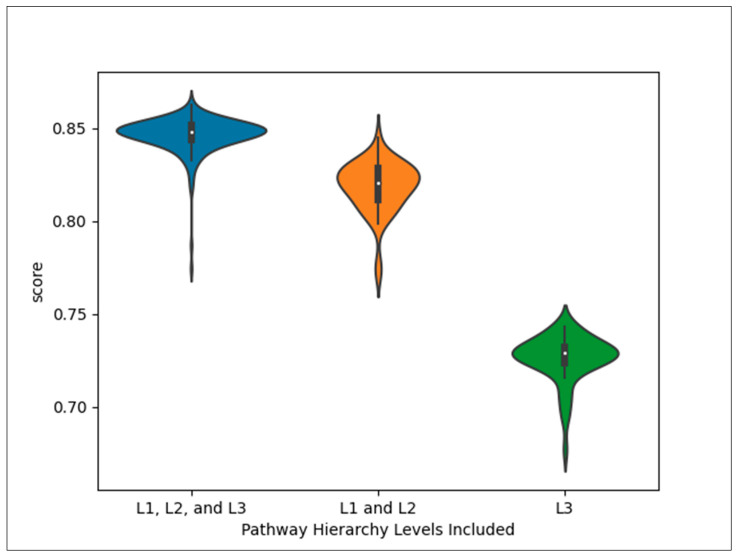
Distribution of MCCs across CV iterations for each dataset.

**Figure 2 metabolites-14-00582-f002:**
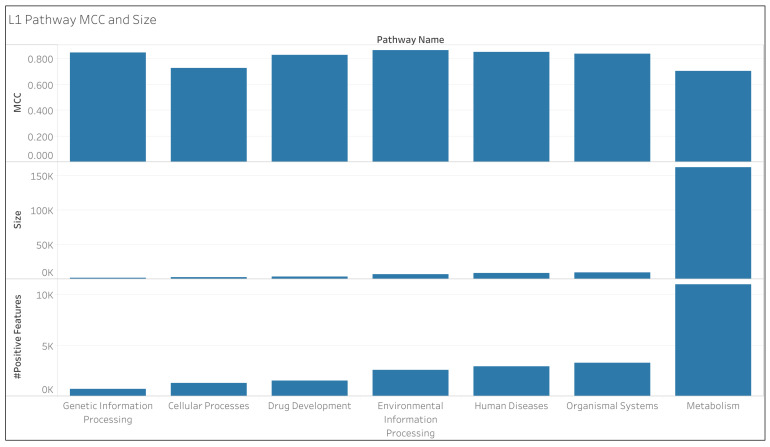
L1 pathway MCC and size as well as the number of pathway features with a positive value.

**Figure 3 metabolites-14-00582-f003:**
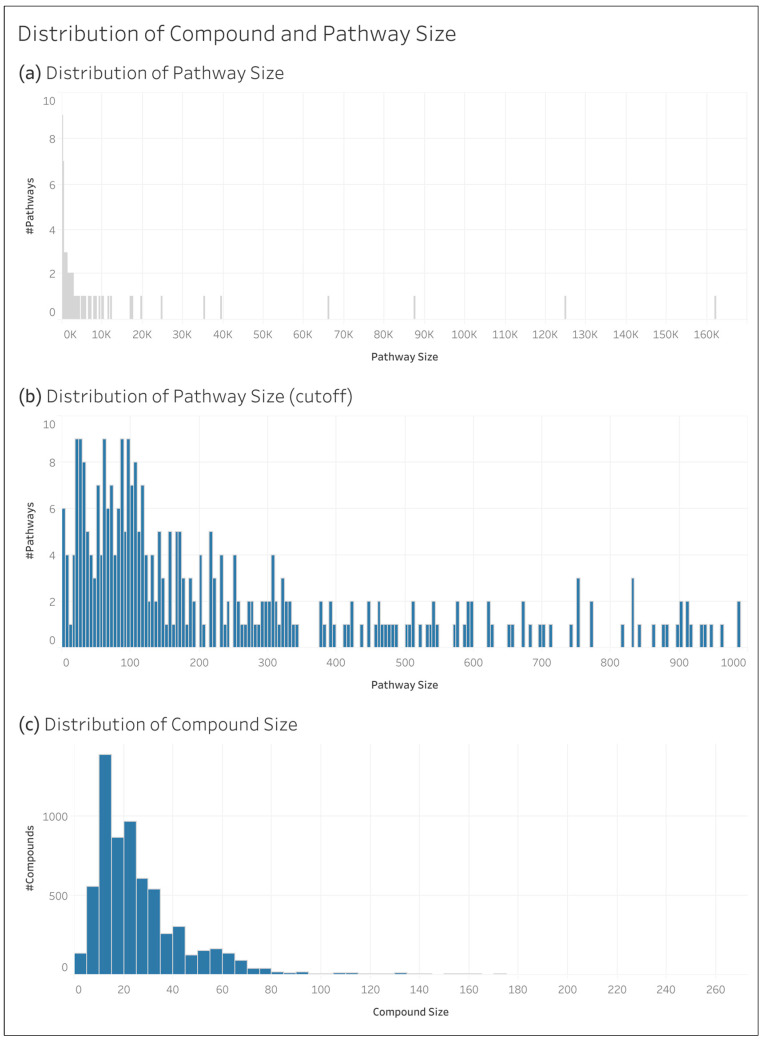
Distribution of pathway and compound size in the full KEGG dataset: (**a**) size distribution of all the pathways; (**b**) distribution of pathways of a size less than 1000; and (**c**) size distribution of the compounds. Size in this context is the number of non-hydrogen atoms in a compound or pathway (summed across the compounds associated with the pathway).

**Figure 4 metabolites-14-00582-f004:**
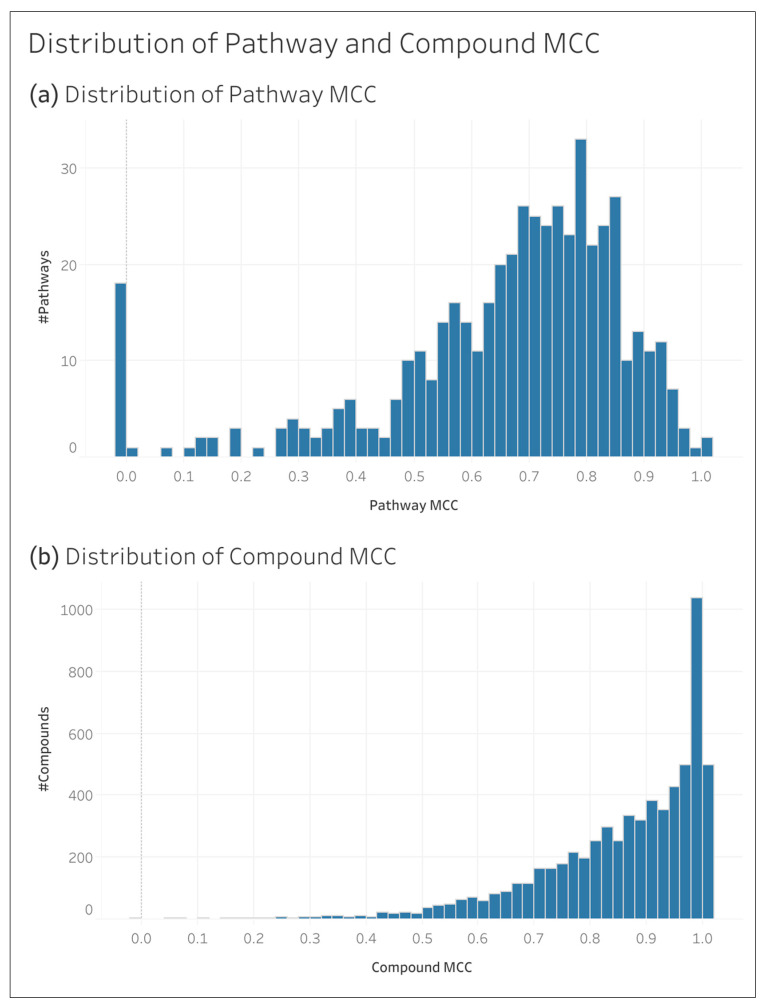
Distribution of the MCCs of individual pathways and compounds in the full KEGG dataset: (**a**) distribution of the pathway MCCs; and (**b**) distribution of the compound MCC.

**Figure 5 metabolites-14-00582-f005:**
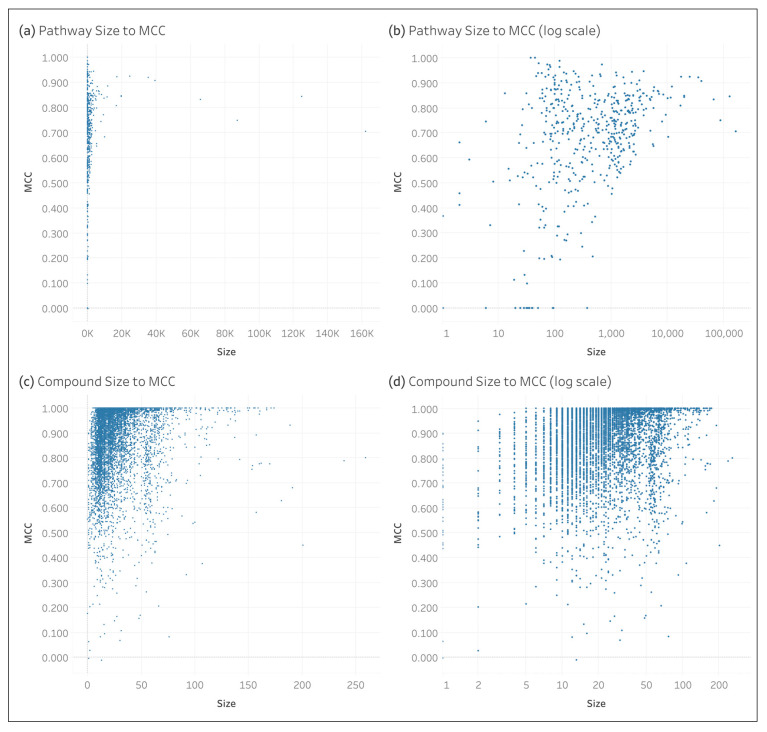
Relation of the pathway and compound size to the individual MCC of the full KEGG dataset: (**a**) pathway size to pathway MCC; (**b**) pathway size to pathway MCC with log scale *x*-axis; (**c**) compound size to compound MCC; and (**d**) compound size to compound MCC with log scale *x*-axis.

**Table 1 metabolites-14-00582-t001:** Correlation results comparing the size filter thresholds to the MCCs of the largest compounds and pathways.

Filter Type	Pearson Correlation Coefficient	*p* Value
**Compound**	0.0107	0.0042
**Pathway**	−0.0336	0.0067

**Table 2 metabolites-14-00582-t002:** Dataset size statistics comparing the dataset with the pathways of all the hierarchy levels to the L2 and L3 pathways only and the L3 pathways only.

Dataset	#Compounds	#Pathways	#Entries
**L1, L2, and L3**	6485	502	**3,255,470**
**L2 and L3**	6485	495	**3,210,075**
**L3**	6485	439	**2,846,915**

**Table 3 metabolites-14-00582-t003:** Dataset size statistics of the full KEGG dataset with all the KEGG pathways and compounds compared to the prior dataset containing only metabolic pathways and metabolites.

Dataset	#Compound Features	#Pathway Features	#Compounds	#Pathways	#Entries	Reference
**Metabolic pathways**	14,655	8977	5683	184	1,045,672	[17]
**Full KEGG**	16,509	11,321	6485	502	**3,255,470**	**Current Study**

**Table 4 metabolites-14-00582-t004:** MCC for all the predictions in each CV iteration by the pathway hierarchy levels included in the dataset.

Pathway Hierarchy Levels Included	Mean MCC	Median MCC	Standard Deviation
**L1, L2, and L3**	0.847	0.848	0.0098
**L2 and L3**	0.819	0.821	0.0135
**L3**	0.726	0.729	0.0127

**Table 5 metabolites-14-00582-t005:** MCCs of the pathways by their hierarchy level for each set of hierarchy levels included in the dataset.

Pathway Hierarchy Levels in Dataset	Pathway Hierarchy Level in Test Set	MCC	TP	TN	FP	FN
**L1, L2, and L3**	L1	0.950	145,888	747,972	8497	4188
	L2	0.904	272,893	6,937,627	35,588	19,742
	L3	0.774	367,169	56,356,625	123,691	89,520
**L2 and L3**	L2	0.894	67,456	1,733,138	9833	5538
	L3	0.769	91,863	14,087,893	32,136	22,543
**L3**	L3	0.726	85,204	14,085,671	34,729	28,996

**Table 6 metabolites-14-00582-t006:** MCCs across all the pathways within each L1 pathway.

Top Pathway	MCC	TP	TN	FP	FN
**Genetic Information Processing**	0.866	2775	1,034,201	585	280
**Metabolism**	0.856	692,950	24,770,413	136,651	89,420
**Environmental Information Processing**	0.802	19,765	3,601,466	5697	4004
**Human Diseases**	0.785	26,662	11,891,608	8012	6576
**Drug Development**	0.748	5366	6,732,952	1570	2042
**Organismal Systems**	0.747	31,386	11,615,945	11,873	9330
**Cellular Processes**	0.728	6798	3,877,363	3347	1789

**Table 7 metabolites-14-00582-t007:** Scores for each L1 pathway.

L1 Pathway	MCC	F1 Score	Specificity	Precision	Recall	TP	TN	FP	FN
**Environmental Information Processing**	0.864	0.871	0.991	0.847	0.897	5853	121,547	1058	675
**Human Diseases**	0.853	0.861	0.991	0.845	0.877	6328	121,432	1162	886
**Genetic Information Processing**	0.848	0.849	0.998	0.828	0.871	1162	128,224	241	172
**Organismal Systems**	0.837	0.847	0.987	0.813	0.884	7034	119,505	1621	921
**Drug Development**	0.827	0.831	0.996	0.820	0.842	2072	126,373	456	389
**Cellular Processes**	0.728	0.731	0.993	0.679	0.793	1870	126,397	885	488
**Metabolism**	0.706	0.985	0.594	0.975	0.995	121,569	4494	3074	657

**Table 8 metabolites-14-00582-t008:** Performance of the model trained on the metabolic KEGG pathways compared to that of all the pathways.

Dataset	Mean MCC	Median MCC	Standard Deviation	Reference
**All L1, L2, and L3**	0.847	0.848	0.0098	Current Study
**All L2 and L3**	0.819	0.821	0.0135	Current Study
**Metabolic L2 and L3**	0.800	-	0.021	[17]
**All L3**	0.726	0.729	0.0127	Current Study
**Metabolic L3**	0.655	-	0.031	[17]

**Table 9 metabolites-14-00582-t009:** Performance of the model trained on the metabolic KEGG pathways compared to that of all the pathways separated by hierarchy level.

Dataset	Pathway Hierarchy Level in Test Set	MCC	Reference
**All L1, L2, and L3**	All L1	0.950	Current Study
	All L2	0.904	
	All L3	0.774	
**All L2 and L3**	All L2	0.894	
	All L3	0.769	
**Metabolic L2 and L3**	Metabolic L2	0.891	[17]
	Metabolic L3	0.726	

## Data Availability

All the code and data for reproducing these analyses can be accessed via the following Figshare item: https://doi.org/10.6084/m9.figshare.27172941 (accessed on 1 June 2024).

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
