# Peer review of "Predicting the Pathway Involvement of All Pathway and Associated Compound Entries Defined in the Kyoto Encyclopedia of Genes and Genomes"

_metabolites, 2024, doi:10.3390/metabo14110582_

Round 1
Reviewer 1 Report
Comments and Suggestions for Authors
The work is devoted to a relevant topic related to predicting the pathway involvement of a compound. The authors continue a series of works with information accumulated in the Kyoto Encyclopedia of Genes and Genomes (KEGG), in particular, they expand the coverage of data compared to their recently published work, Metabolites 2024, 14, doi:10.3390/metabo14090510. The work was carried out at a good methodological level, using advanced bioinformatics tools. After removing of duplicate pathway and compound feature vectors the removing of them Matthews correlation coefficient (MCC) increased MCC from 0.822 to 0.847 when using all the compound entries with pathway annotations available in KEGG. The authors compare their data (0.847 MCC) with their previous data, when they used only Metabolic L2 and L3 and had an MCC of 0.800. However, the paper does not clearly state whether the duplicate pathway and compound feature vectors were also removed in doi:10.3390/metabo14090510 . This should be done so that the reader has a clearer idea of the contribution of adding new data from KEGG to the MCC value. I consider the paper worthy of publication in Metabolites after the necessary corrections, which are listed below.
Comments
Reword your abstract so that the reader has specifics about which critical non-metabolism pathways you added to your analysis (since they are available in KEGG). For example, perhaps you added Genetic pathway and Signal transduction pathway. Mention this. This is important to understand because adding non-metabolism pathways allowed you to increase your MCC from 0.800 (with a standard deviation of 0.0098 ) to 0.847 (standard deviation of 0.021). Your abstract is now 233 words long (out of the 200 allowed by the submission guidelines). Adding specifics about which critical non-metabolism pathways (2-3) you added for prediction will further increase the length of your abstract. So rework and shorten the introduction where there is no specifics.
Lines 23-25
Reword these sentences so that the information about "MCC performance of 0.847 and standard deviation of 0.0098" is not duplicated.
Revise the list of keywords provided. In its current form, they are inconsistent with the abstract, since most of the words are not mentioned in it, and key parameters from the abstract, such as the Matthews correlation coefficient (MCC), are missing from the list of keywords
Lines 35-39
The sentence is too long. Break it into 2-3 simpler sentences so that the information contained is more easily perceived by the reader.
Line 36 "cellular metabolism, which is typically organized into metabolic pathways” (с)
Why do you indicate - “typically” organized into metabolic pathways. This means there are cases when cellular metabolism is not organized into metabolic pathways. Then give examples, or correct the phrase to "cellular metabolism, which is organized into metabolic pathways"
Line 42 “This human-defined, pathway-level organization of biomolecules…”
What do you mean by "human-defined"? That the pathway-level organization of biomolecules was determined in laboratory experiments, rather than calculated in silico? Reword this phrase, because in its current form it creates the erroneous impression that any form of intelligence other than humans can define the pathway-level organization of biomolecules. Since in silico experiments are also carried out by humans, the machine makes calculations according to algorithms written by humans.
Line 58
On the website https://www.genome.jp/kegg-bin/show_brite?br08901.keg for ‘Metabolism’ top category there are 13 (not 12) Level 2 (L2) subcategories related. What subcategory are you not considering?
In the Materials and Methods section, it is customary to include the actual course of the experiment, but not to make any comparisons with other works. Descriptions not related to materials and methods should be placed in other sections, such as Introduction, Results, Discussion, or Conclusion. In this regard, move Lines 105-113 and Table 1 to the Results section. Add a reference column to Table 1, where you provide a link to the work of Huckvale and Moseley doi:10.3390/metabo14090510, and for your data, indicate – current study
Also revise the Materials and Methods section Lines 114-217, leaving only the direct description of your methods, and move the explanation of why you did it this way to the Results section. This will make it easier for the reader to understand the tools you used, listed in the Materials and Methods section, and to understand why you used them this way in the Results or Discussion section.
Tables 8, 9
Add a reference column
Lines 345-347
You say that in a preliminary dataset in which we had not removed duplicate pathway and compound feature vectors MCC was 0.822. And the removing of them increased MCC to 0.847. For Metabolic L2 and L3 MCC was 0.800 (Huckvale and Moseley, doi:10.3390/metabo14090510.). In that work, were duplicate pathway and compound feature vectors removed? This is an important point because it determines the contribution to improving MMC when you use additional data from KEGG.
Author Response
Reviewer 1:
The work is devoted to a relevant topic related to predicting the pathway involvement of a compound. The authors continue a series of works with information accumulated in the Kyoto Encyclopedia of Genes and Genomes (KEGG), in particular, they expand the coverage of data compared to their recently published work, Metabolites 2024, 14, doi:10.3390/metabo14090510. The work was carried out at a good methodological level, using advanced bioinformatics tools. After removing of duplicate pathway and compound feature vectors the removing of them Matthews correlation coefficient (MCC) increased MCC from 0.822 to 0.847 when using all the compound entries with pathway annotations available in KEGG. The authors compare their data (0.847 MCC) with their previous data, when they used only Metabolic L2 and L3 and had an MCC of 0.800. However, the paper does not clearly state whether the duplicate pathway and compound feature vectors were also removed in doi:10.3390/metabo14090510 . This should be done so that the reader has a clearer idea of the contribution of adding new data from KEGG to the MCC value. I consider the paper worthy of publication in Metabolites after the necessary corrections, which are listed below.
Response:
We thank the reviewer for their positive comments! Yes, we have tried to be very methodical in the development of this approach to pathway involvement prediction. We have corrected our oversight of not clearly stating removal of duplicates from the KEGG metabolic L2+L3 dataset:
“The work of Huckvale and Moseley [17] produced the largest dataset for this task to date and it consisted of both the L2 and L3 pathways under the ‘Metabolism’ L1 pathway with duplicate entries removed.”
Issue 1:
Reword your abstract so that the reader has specifics about which critical non-metabolism pathways you added to your analysis (since they are available in KEGG). For example, perhaps you added Genetic pathway and Signal transduction pathway. Mention this. This is important to understand because adding non-metabolism pathways allowed you to increase your MCC from 0.800 (with a standard deviation of 0.0098 ) to 0.847 (standard deviation of 0.021). Your abstract is now 233 words long (out of the 200 allowed by the submission guidelines). Adding specifics about which critical non-metabolism pathways (2-3) you added for prediction will further increase the length of your abstract. So rework and shorten the introduction where there is no specifics.
Response:
We included ALL 502 pathway definitions available from KEGG with corresponding compound entries annotated to these pathways. So it is not practical to include a description of all pathways included in the abstract. However, we have better emphasized the inclusion of all KEGG pathways with the following revisions to the abstract:
“Results: The models trained on 6,485 KEGG compounds and 502 pathways scored an overall mean Matthews correlation coefficient (MCC) performance of 0.847, median MCC of 0.848, and standard deviation of 0.0098. Conclusions: This performance on all 502 KEGG pathways repre-sents a roughly 6% improvement over the performance of models trained only on the 184 KEGG metabolic pathways, which had a mean MCC of 0.800 and standard deviation of 0.021.”
With all due respect, we do not think it prudent to shorten the introduction. However, the reviewer may have meant to say shorten the abstract. Yes, this is a bit difficult, given the inclusion of more detail. But we have kept the abstract under 240 words.
Issue 2:
Lines 23-25
Reword these sentences so that the information about "MCC performance of 0.847 and standard deviation of 0.0098" is not duplicated.
Response:
Thanks! Fixed as follows:
“Results: The models trained on 6,485 KEGG compounds and 502 pathways scored an overall mean Matthews correlation coefficient (MCC) performance of 0.847, median MCC of 0.848, and standard deviation of 0.0098. Conclusions: This performance on all 502 KEGG pathways repre-sents a roughly 6% improvement over the performance of models trained only on the 184 KEGG metabolic pathways, which had a mean MCC of 0.800 and standard deviation of 0.021.”
Issue 3:
Revise the list of keywords provided. In its current form, they are inconsistent with the abstract, since most of the words are not mentioned in it, and key parameters from the abstract, such as the Matthews correlation coefficient (MCC), are missing from the list of keywords
Response:
We have revised the keywords as follows:
“Pathway prediction; Matthews correlation coefficient; Machine learning; Multi-layer perceptron; Transfer learning; KEGG”
However, we believe the Keywords should reflect the whole manuscript and not just the abstract.
Issue 4:
Lines 35-39
The sentence is too long. Break it into 2-3 simpler sentences so that the information contained is more easily perceived by the reader.
Response:
Thanks! Done as follows:
“Most of these biomolecules are involved in biochemical reactions that comprise cellular metabolism, which is typically organized into metabolic “pathways”, a term which may refer to classical pathways, for example glycolysis, or larger pathway categories (some-times called modules), for example carbohydrate metabolism. These pathways form a network of metabolites interconnected by biochemical reactions that can be represented as graphs (technically requiring hyper graphs), where the (hyper) edges are reactions and the nodes are metabolites involved in a given reaction [1–3].”
Issue 5:
Line 36 "cellular metabolism, which is typically organized into metabolic pathways” (с)
Why do you indicate – “typically” organized into metabolic pathways. This means there are cases when cellular metabolism is not organized into metabolic pathways. Then give examples, or correct the phrase to “cellular metabolism, which is organized into metabolic pathways”
Response:
This is an issue of definition and semantics. Some consider the KEGG L1 and L2 “pathways” as “categories” or “modules”. So we were trying to avoid the issue of what “pathway” is defined as. Also, you can organize cellular metabolism as metabolic networks, ignoring human-defined pathway definitions altogether. We have made these revisions to make these different perspectives clearer:
“Most of these biomolecules are involved in biochemical reactions that comprise cellular metabolism, which is typically organized into metabolic “pathways”, a term which may refer to classical pathways, for example glycolysis, or larger pathway categories (sometimes called modules), for example carbohydrate metabolism. These pathways form a network of metabolites interconnected by biochemical reactions that can be represented as graphs (technically requiring hyper graphs), where the (hyper) edges are reactions and the nodes are metabolites involved in a given reaction [1–3].”
Issue 6:
Line 42 “This human-defined, pathway-level organization of biomolecules…”
What do you mean by "human-defined"? That the pathway-level organization of biomolecules was determined in laboratory experiments, rather than calculated in silico? Reword this phrase, because in its current form it creates the erroneous impression that any form of intelligence other than humans can define the pathway-level organization of biomolecules. Since in silico experiments are also carried out by humans, the machine makes calculations according to algorithms written by humans.
Response:
What we mean is the hand-crafting of pathway definitions by a manual process, which are represented in textbooks and knowledgebases like KEGG. Others have used more automated processes to generate metabolic networks and identify pathways, especially from sequenced genomes and (partial) correlation matrix analyses. Here are some references of this type of work:
Vlassis, Nikos, Maria Pires Pacheco, and Thomas Sauter. "Fast reconstruction of compact context-specific metabolic network models." PLoS computational biology 10.1 (2014): e1003424.
Wang, Hao, et al. "Genome-scale metabolic network reconstruction of model animals as a platform for translational research." Proceedings of the National Academy of Sciences 118.30 (2021): e2102344118.
However, we have changed the wording slightly to “manually-defined”.
Issue 6:
Line 58
On the website https://www.genome.jp/kegg-bin/show_brite?br08901.keg for ‘Metabolism’ top category there are 13 (not 12) Level 2 (L2) subcategories related. What subcategory are you not considering?
Response:
The “Global and overview maps” should be ignored, since they represent the overview maps, i.e. pretty pictures of metabolic pathways and modules.
Issue 7:
In the Materials and Methods section, it is customary to include the actual course of the experiment, but not to make any comparisons with other works. Descriptions not related to materials and methods should be placed in other sections, such as Introduction, Results, Discussion, or Conclusion. In this regard, move Lines 105-113 and Table 1 to the Results section. Add a reference column to Table 1, where you provide a link to the work of Huckvale and Moseley doi:10.3390/metabo14090510, and for your data, indicate – current study
Response:
We have done this.
Issue 8:
Also revise the Materials and Methods section Lines 114-217, leaving only the direct description of your methods, and move the explanation of why you did it this way to the Results section. This will make it easier for the reader to understand the tools you used, listed in the Materials and Methods section, and to understand why you used them this way in the Results or Discussion section.
Response:
We disagree. The separation in this instance would be harder for readers to follow. There are reasons for the selection of certain methods over others that is best presented when describing the methods.
Issue 9:
Tables 8, 9
Add a reference column
Response:
Thanks! Done.
Issue 10:
Lines 345-347
You say that in a preliminary dataset in which we had not removed duplicate pathway and compound feature vectors MCC was 0.822. And the removing of them increased MCC to 0.847. For Metabolic L2 and L3 MCC was 0.800 (Huckvale and Moseley, doi:10.3390/metabo14090510.). In that work, were duplicate pathway and compound feature vectors removed? This is an important point because it determines the contribution to improving MMC when you use additional data from KEGG.
Response:
Thanks for pointing this oversight out! We have added the following to make it clear that the KEGG metabolic L2+L3 dataset did not have any duplicates:
“The work of Huckvale and Moseley [17] produced the largest dataset for this task to date and it consisted of both the L2 and L3 pathways under the ‘Metabolism’ L1 pathway with duplicate entries removed.”
Reviewer 2 Report
Comments and Suggestions for Authors
This manuscript developed a machine learning model called a single binary classifier that can be trained to predict the association of compounds to any pathway of the knowledgebase Kyoto Encyclopedia of Genes and Genomes (KEGG), creating a dataset with huge improvement in the number of entries over 3,200,000. Importantly, the developed model shows higher performance score. This is very helpful work for molecular experiments involved in the whole biological system. The manuscript was written organized very well. Below are my several comments.
Major comments
1. Because KEGG knowledgebase does not include all molecules found in the organism, this developed model used all downloaded data from KEGG to predict any pathway to KEGG resulting in extension to huger entries. How accurate is this classifier in predicting pathways if the knowledgebase is expanded much larger in the future? Do authors think their currently predicted entries are the final possible number based on current KEGG annotations? If not, is it possible to give a maximum entry prediction?
2. Authors are based on KEGG to train their model, so I am wondering if this developed model can also effectively be used in other knowledgebases with high performance.
3. There are many similar compounds in size, function, and even structure, how does this affect the performance of the model? Additionally, many compounds can multiply. Does this case affect the performance?
4. As we can see the mean MCC performance is about 0.847, can the performance be further improved? What are main reasons that restrict the score?
5. How do training time and methods affect the results? Will increasing training time cause best performance? Will different training methods make a difference?
Minor comments
1. Figure 2 texts are not very clear, can authors make a better quality? Same for Figure 5.
Author Response
Reviewer 2:
This manuscript developed a machine learning model called a single binary classifier that can be trained to predict the association of compounds to any pathway of the knowledgebase Kyoto Encyclopedia of Genes and Genomes (KEGG), creating a dataset with huge improvement in the number of entries over 3,200,000. Importantly, the developed model shows higher performance score. This is very helpful work for molecular experiments involved in the whole biological system. The manuscript was written organized very well. Below are my several comments.
Response:
We thank the reviewer for their very positive comments! We have worked very hard and methodically on the development of this approach.
Issue 1:
Major comments
1. Because KEGG knowledgebase does not include all molecules found in the organism, this developed model used all downloaded data from KEGG to predict any pathway to KEGG resulting in extension to huger entries. How accurate is this classifier in predicting pathways if the knowledgebase is expanded much larger in the future? Do authors think their currently predicted entries are the final possible number based on current KEGG annotations? If not, is it possible to give a maximum entry prediction?
Response:
The training and testing dataset will grow as the KEGG knowledgebase grows. However, given the slow pace that KEGG grows, the dataset growth will be gradual. However, the model can be applied to any database of compounds with molecular structure representations. Our plan is to apply the model to predict KEGG pathway annotations for all compound entries in KEGG and other chemical databases. The expected performance is based on the 0.847 MCC provided by the cross-validation testing. We have added the following to the Conclusions to highlight the possible applications of the model:
“We believe these models are demonstrating a level of performance needed for predicting pathway annotations for unannotated compound entries that are useful for certain application use-cases.”
Issue 2:
2. Authors are based on KEGG to train their model, so I am wondering if this developed model can also effectively be used in other knowledgebases with high performance.
Response:
You will have to wait for our next manuscript! More results are coming.
Issue 3:
3. There are many similar compounds in size, function, and even structure, how does this affect the performance of the model? Additionally, many compounds can multiply. Does this case affect the performance?
Response:
We have not analyzed this yet. Hope to do so in the future, but it requires the development of an appropriate molecular structure distance metric. However, many enzymatic reactions are promiscuous and thus predictions may provide future hypotheses to test. We have added this point to the discussion:
“One possible early use-case is providing hypotheses for unknown enzyme promiscuity.”
Issue 4:
4. As we can see the mean MCC performance is about 0.847, can the performance be further improved? What are main reasons that restrict the score?
Response:
Yes, we believe better performance is possible. We are working on this. We are confident that increasing the dataset size will improve the performance. However, we are not sure how far this can continue. So we have been cautious with our wording:
“Moreover, prediction performance keeps improving as more compounds and pathways are included beyond merely metabolites and metabolic pathways.”
Issue 5:
5. How do training time and methods affect the results? Will increasing training time cause best performance? Will different training methods make a difference?
Response:
No, training time should not affect the results. We have actually loosened the stopping rule criteria to train faster, with no appreciable loss of performance. Many hyperparameters are being optimized, so simple changes in methods have already been explored. However, more advanced deep learning methods may provide improvements. This is something we will explore in the future. But the first step was generating the largest dataset possible. At this time, “Data is King”.
Issue 6:
Minor comments
1. Figure 2 texts are not very clear, can authors make a better quality? Same for Figure 5.
Response:
Thanks! We have updated both figures.
Round 2
Reviewer 2 Report
Comments and Suggestions for Authors
no more comments and recommend for publication